# Surface wetting is a key determinant of α-synuclein condensate maturation

Rebecca J. Thrush[1,2], Devkee M. Vadukul[1], Siân C. Allerton[1,2], Marko Storch[3,4] & Francesco A. Aprile [1,2] ✉

α-Synuclein condensates can mature into amyloid fibrils, demonstrating a link between phase separation and amyloid aggregation. However, the mechanisms driving this maturation are not fully understood, particularly in the context of pathological post-translational modifications that modulate α-synuclein amyloid aggregation. Although often studied in isolation, condensates appear to interact with surfaces in vitro and in the cell. Notably, the N-terminus of α-synuclein is implicated in membrane binding and may influence condensate-surface interactions. Here, we developed a microscopy-based protocol to investigate how N-terminal truncation affects α-synuclein condensate formation, surface wetting, and maturation. We found that N-terminal truncation enhances condensate wettability and accelerates maturation. Conversely, perturbing condensate-surface interactions reduces condensate wettability and delays maturation. These results suggest that enhanced wettability promotes condensate maturation, likely by increasing condensate surface-to-volume ratios. Our findings reveal distinct mechanistic roles for the N-terminus of α-synuclein and highlight condensate wettability and interfacial dynamics as key modulators of aggregate formation.

The intrinsically disordered protein (IDP) α-synuclein (α-syn) exists as a monomer and multimers in neurons, where it mediates neurotransmitter release and synaptic function[1–6]. α-Syn is also found as insoluble amyloid fibrils with a characteristic cross-β-sheet rich core structure[7,8]. The aberrant accumulation of α-syn amyloid fibrils is the major hallmark of neurodegenerative diseases, including Parkinson's disease (PD), dementia with Lewy bodies, and multiple system atrophy[9,10].

Many post-translational modifications (PTMs) affect α-syn amyloid aggregation and PD pathology[11–20]. These PTMs include acetylation, phosphorylation, and terminal domain truncation[12–15]. One PTM that remains largely uncharacterized is N-terminal truncation, which has been observed both in vitro[21,22] and in vivo[13–15,23–26]. The N-terminus of α-syn plays key roles in both the physiological and pathological functions of the protein, e.g., the domain acts as a membrane anchor region[27,28], modulating synaptic vesicle fusion to the presynaptic terminal[4,29]. Yet, in vitro evidence shows that vesicle-bound α-syn is often predisposed to self-association and can nucleate free monomers to further promote aggregation[27,30]. The N-terminus also plays a role in the growth phase of amyloid aggregation, specifically elongation and secondary nucleation[31,32]. Thus, unsurprisingly, N-terminal truncations and deletions have been shown to alter α-syn aggregation kinetics, amyloid morphology and stability, membrane binding, and toxicity, relative to the full-length (FL) protein[33–38].

Emerging evidence demonstrates that α-syn can undergo phase separation, resulting in the formation of biomolecular condensates[39–45]. At the early stages of maturation, the condensates exhibit liquid-like properties, allowing α-syn to diffuse within them and exchange with the surrounding environment[39,44]. These liquid-like properties are likely facilitated by weak multivalent interactions between α-syn molecules[46,47]. The condensates can also grow via fusion events (coalescence) and Ostwald ripening[39,44]. With time, the condensates can mature into a solid-like gel, where composing α-syn molecules exhibit reduced mobility[39,44]. This liquid-to-solid transition ultimately leads to the formation of amyloids[39,43,44]. Several environmental conditions can promote α-syn phase separation. A molecular crowding agent, commonly poly-ethylene glycol (PEG), is often necessary, although evaporation, poly-cations, metal ions or other amyloidogenic proteins, e.g., tau, can also induce phase separation[39–44]. Interestingly, recent evidence indicates that, under certain conditions, the condensate-bulk solution interface can act as a catalyst for the aggregation of amyloidogenic proteins, including α-syn[45,48,49].

The exact protein regions and interactions regulating α-syn phase separation remain largely unknown. However, the critical protein concentration required for phase separation is significantly reduced by lowering pH, increasing salt, or introducing metal ions[39,40]. Similarly, interaction of the negative C-terminus of α-syn with positively charged regions of tau or poly-L-lysine (PLK) induces complex coacervation; the co-phase separation of two opposingly charged molecules[42]. Moreover, a variant possessing only the core residues 30–110 and C-terminally truncated α-syn both undergo enhanced phase separation and aggregation, relative to FL α-syn[39,41].

[1]Department of Chemistry, Molecular Sciences Research Hub, Imperial College London, London, UK. [2]Institute of Chemical Biology, Molecular Sciences Research Hub, Imperial College London, London, UK. [3]London Biofoundry, Translation and Innovation Hub, Imperial College London, London, UK. [4]Department of Infectious Disease, South Kensington, Imperial College London, London, UK. ✉e-mail: f.aprile@imperial.ac.uk

Together, these findings indicate that disruption of electrostatic interactions involving the terminal domains is a dominant promoter of α-syn condensate formation. However, β-syn, which lacks the hydrophobic eight-residue stretch located in the non-amyloid-β component (NAC) domain of α-syn, does not phase separate, indicating that this domain is critical for condensate formation. Thus, all three domains are key to determining the likelihood of α-syn phase separation and their precise roles in this process must be examined further.

Amyloid aggregation is significantly affected by conditions that promote phase separation. For several proteins, including α-syn, tau and heterogenous nuclear ribonucleoprotein A1 (hnRNPA1), amyloid aggregation is accelerated under phase separation conditions[39,46,48,50,51]. This effect is likely due to increased local protein concentration and enhanced nucleation. Furthermore, various disease factors known to promote α-syn aggregation in the absence of phase separation (e.g., $Cu^{2+}$ and $Fe^{3+}$) also accelerate the onset of phase separation and the subsequent liquid-to-solid transition[39]. However, conditions that promote phase separation can also inhibit amyloid aggregation, as observed for the 42-residue-long amyloid-β peptide, which can be sequestered and stabilized within condensates formed from the low complexity domains of the DEAD-box proteins[52].

The ability of biomolecular condensates to wet surfaces is believed to be a biologically relevant mechanism[53,54]. In cells, condensate wetting on biological surfaces, such as cytoskeletal fibers and lipid membranes, can mediate the structure and function of both the condensate and the surface. This includes the regulation of amyloidogenic protein condensate formation and maturation on lipid surfaces[55,56]. However, the consequences of surface wetting on α-syn condensate stability and cellular toxicity remain unknown.

Despite the importance of the N-terminus, molecular details of how specific N-terminal regions regulate the nucleation and propagation mechanisms of α-syn self-assembly are still unclear. In this study, we used a microscopy-based approach to identify sequence determinants within the N-terminus that regulate the interplay between the surface wettability and maturation of α-syn condensates.

## Results

### Design of the N-terminally truncated α-syn variants
Several N-terminal truncations have been associated with PD. We selected three different truncations (Supplementary Table 1): 5-140, isolated from PD brains and following in vitro incubation[13,22], 11-140, found in Lund human mesencephalic cells[15], and 19-140, found in human appendices[14]. The N-terminal region of these variants differs from the FL protein in both charge and hydrophobicity. 5-140 α-syn has an increased relative charge (ΔCharge) of +1, while both 11-140 and 19-140 α-syn have a decreased ΔCharge of −1, relative to FL α-syn. In terms of N-terminal hydrophobicity, FL α-syn is the most hydrophobic, followed by 11-140 and then 5-140 α-syn. 19-140 α-syn has the lowest hydrophobicity. We recombinantly expressed and purified all three variants, alongside FL α-syn, as confirmed by ESI-MS (Supplementary Fig. 1). Using far-UV CD spectroscopy we found that, like FL α-syn, all truncated variants consist primarily of random coil secondary structure (Supplementary Fig. 2).

### N-terminal truncation does not affect the ability of α-syn to phase separate
α-Syn has been reported to naturally interact with basic proteins, such as synapsins[42,57]. In vitro, α-syn can phase separate when incubated with the polycation PLK and a molecular crowding agent, e.g., PEG, mimicking physiological conditions[42]. To investigate how N-terminal truncation affects α-syn phase separation, we generated phase diagrams for our α-syn variants. We incubated varying concentrations of each protein with 10% PEG and different concentrations of PLK for 15 min, when the highest number of condensates was observed (Fig. 1a, b). Then, we imaged the bulk solution of each sample using differential interference contrast (DIC) microscopy, designed a General Analysis 3 (GA3) recipe that estimates condensate number and area (Supplementary Fig. 3), and plotted the number of

condensates in solution as a function of both α-syn and PLK concentration (Fig. 1a and Supplementary Fig. 4). These values were combined over 3 z-stack images acquired in solution per sample (Fig. 1a). The distance between the planes (200 μm) was chosen to be ~ 100-fold greater than the average condensate diameter (~ 2.5 μm) to ensure no condensate is captured in more than one image.

FL α-syn formed condensates at either 40 or 60 μM with 25 μM PLK, and at 60 μM with 50 μM PLK (Supplementary Figs. 4 and 5). To verify that condensates contained FL α-syn, we performed fluorescence microscopy on a sample of 60 μM FL α-syn, 1% of which was rhodamine labeled, and 25 μM PLK. Fluorescence emission was specifically localized within the condensates, confirming the presence of α-syn (Supplementary Fig. 6). We repeated this analysis on 5-140, 11-140 and 19-140 α-syn. We found that all truncated variants formed condensates, with a similar dependency on the ratio of protein:PLK as FL α-syn (Supplementary Figs. 4 and 7–9). This suggests that residues 1–18 are not required for condensate formation, consistent with existing evidence that this complex coacervation mechanism requires interaction between PLK and the C-terminus of α-syn, and that residues 30–110 are sufficient for α-syn phase separation[39,42].

### N-terminal truncation accelerates α-syn amyloid formation under phase separation conditions
We next investigated whether N-terminal truncation affects the time-evolution of α-syn condensates. We prepared solutions containing 60 μM α-syn, 10% PEG, and 25 μM PLK, as we observed significant condensate formation under this condition for all α-syn variants (Supplementary Fig. 4). We monitored condensate maturation in the bulk of the solution using DIC microscopy and then quantified condensate area and number variation with time (Fig. 1a). Additionally, we imaged the bottom surface of the well at each time point to monitor any condensate or aggregate sedimentation during incubation (Fig. 1a).

First, we examined FL α-syn and found that condensates increased in size (from ~ 1 to ~ 5 μm² area) during the first ~ 3 h of incubation (Fig. 1b, c and Supplementary Fig. 10a). This increase in size was attributed in part to coalescence, which was detected after ~ 20 min, and evidences the liquid-like nature of early α-syn condensates (Supplementary Fig. 11). After ~ 3 h, the condensates stopped growing (Fig. 1b, c and Supplementary Fig. 10a), possibly because their low number reduced the likelihood of proximity-dependent growth events (i.e., coalescence and Ostwald ripening). Alternatively, the condensates may have transitioned into a gel-like phase, preventing further growth[39,44].

In contrast, the number of condensates in the bulk solution decreased continuously after 15 min (Fig. 1b, d and Supplementary Fig. 10b). By 20 h, very few condensates were visible in solution. This behavior was partially attributed to coalescence, which would decrease the number of condensates. We also imaged the bottom of the well throughout incubation and found a progressive increase in the number of condensates wetting the well surface (Fig. 1e), indicating sedimentation also contributes to the decay in condensate number in solution. When manually imaged at 1.5 h we observed fusion events between condensates in solution and surface-wetted condensates (Supplementary Fig. 12), explaining the progressive increase in surface-wetted condensate size over time (Fig. 1e). We did not observe condensates, surface wetting, or aggregates in any of the control samples, i.e., 60 μM FL α-syn alone, 25 μM PLK alone or phase separation buffer (Supplementary Figs. 13 and 14).

To investigate the transition of FL α-syn condensates into amyloids, we performed fluorescence emission measurements in a microplate reader using the amyloid-sensitive probe Thioflavin-T (ThT). We prepared samples containing 60 μM α-syn under the same phase separation-inducing conditions as above. Then, we recorded the ThT fluorescence intensity over time as a readout of amyloid formation. To mimic the experimental conditions of the microscopy measurements, we took readings at the well center every 20 min with 30 flashes per well, minimizing sample agitation and overheating from the excitation light. Aggregation commenced after ~ 20 h (Fig. 1d). In the absence of PLK, we did not observe any significant

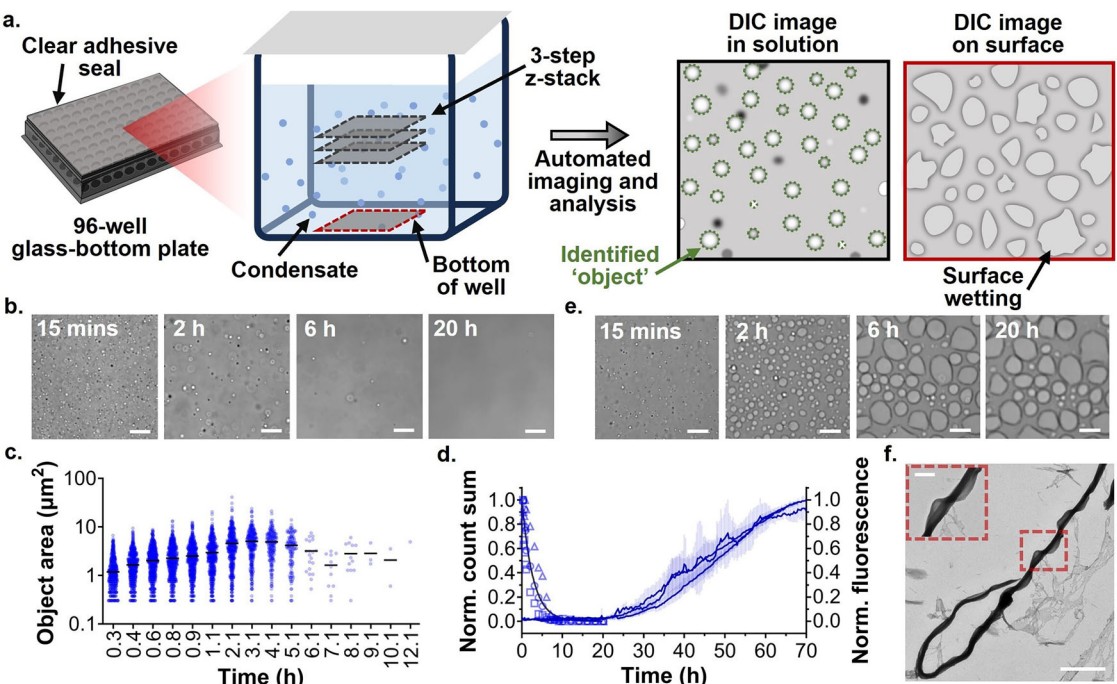

**Fig. 1 | Condensate and amyloid formation of FL α-syn under phase separation conditions. a** A diagram outlining the methodology used to monitor α-syn liquid condensate formation, sedimentation and maturation. Samples, loaded in a 96-well glass-bottom plate sealed with a clear film, are incubated at 37 °C. Automated 3-step z-stack imaging in solution, alongside bottom-of-well imaging, is performed at selected time points over 20 h. Solution image analysis is then performed to quantify the number and area of all in focus objects within the image. **b** Representative DIC images acquired in solution during FL α-syn incubation. Scale bars represent 25 μm. **c** DIC microscopy-derived object area distribution over time. Each timepoint was compiled over three z-stack images. Mean areas are indicated by a solid black line.

**d** Normalized total object count (left Y-axis) and normalized ThT intensity (right Y-axis) as a function of time. Three biological repeats are shown for the normalized total object count data (circles, squares, or triangles), which are globally fitted using a one-phase decay model (black line). Three biological repeats are shown for the normalized ThT aggregation data (blue lines), where each biological repeat is the mean of three technical replicates and error bars represent the standard deviation of the mean. **e** Representative DIC images acquired at the bottom of the well during FL α-syn incubation. Scale bars represent 25 μm. **f** Representative TEM image, with magnification, of an aliquot collected at the endpoint of a phase separation ThT aggregation assay. Scale bars represent 2 μm and 500 nm, respectively.

aggregation within the time frame of this experiment (Supplementary Fig. 15a). These data are consistent with our microscopy observations (Fig. 1b, d, Supplementary Figs. 13 and 14) and existing evidence that phase separation can accelerate amyloid aggregation[39,46,48,50,51].

To characterize the aggregates formed following phase separation, we analyzed the endpoint of this assay. Microscopy revealed ThT-positive aggregates, including large objects with protruding fibrils (Supplementary Fig. 16 and Supplementary Movie 1). TEM also confirmed the presence of large, spiraled fibrils (Fig. 1f). To understand the composition and secondary structure of these aggregates, we isolated and analyzed the insoluble protein fraction. We found that this fraction contained α-syn and displayed β-sheet secondary structure, as assessed by dot-blot and far-UV CD spectroscopy, respectively (Supplementary Fig. 17). Altogether, these data indicate that amyloid-like aggregates are formed under phase separation conditions. These amyloids may consist of mixed fibrils in which PLK, which is unlikely to reside in the amyloid core due to its high charge, may bind the negatively charged C-terminal fuzzy coat and promote lateral association, resulting in the thick fibrils we observed. In the absence of PLK, we observed a small number of thin fibrils by TEM, consistent with the ThT intensity data (Supplementary Fig. 18). To further characterize the FL fibrils formed via phase separation, we evaluated their seeding capacity and found that FL seeds efficiently promoted aggregation, yielding kinetics with a short lag phase indicative of an activation step before steady elongation (Supplementary Fig. 19).

Next, we repeated the DIC microscopy analysis with the truncated α-syn variants. In the case of 5-140 α-syn, the time evolution of condensates in the bulk was similar to FL α-syn (Fig. 2a and Supplementary Figs. 20–22). In contrast, 11-140 and 19-140 α-syn condensates grew more slowly but persisted for longer (Fig. 2b, c, Supplementary Figs. 20–22). This behavior

may be a consequence of altered sedimentation dynamics. In fact, increasing truncation reduces N-terminal hydrophobicity (Supplementary Table 1), which may enhance water retention within condensates, lowering their density and thereby slowing sedimentation.

DIC microscopy of 5-140 α-syn revealed non-spherical objects in the bulk solution within ~ 2 h, indicating the formation of small solid aggregates (Supplementary Fig. 22). By 20 h, bottom-of-well imaging showed a large fibrillar network (Fig. 2a and Supplementary Fig. 23), and TEM at the end of incubation confirmed long, thick fibril clumps (Fig. 2b).

In contrast, 11-140 and 19-140 α-syn showed extensive surface wetting at 20 h, comparable to the FL protein (Fig. 2a). Both truncations resulted in a greater surface area of wetted condensates, which also contained small aggregates (Fig. 2a and Supplementary Fig. 23). The morphology of these condensates was less spherical, and their borders had substantially lower contrast (seen at 20 h in Supplementary Fig. 23), indicating a thinner condensate layer. This suggests condensates of 11-140 and 19-140 α-syn have a lower contact angle on the surface and thus increased wettability, relative to FL α-syn.

Finally, we examined amyloid formation under phase separation-promoting conditions using the ThT intensity assay. We observed faster aggregation kinetics for all truncated variants, relative to FL α-syn (Fig. 3a and Supplementary Fig. 15b–d). DIC, ThT fluorescence and electron microscopy at the endpoint of aggregation revealed extensive thick fibril networks for all variants (Fig. 3b and Supplementary Movies 2–4). The insoluble protein fractions were α-syn positive and contained β-sheet secondary structure, indicative of amyloids (Supplementary Fig. 17). When truncated α-syn was incubated in the absence of PLK, we did not observe condensates, surface wetting, or aggregates within 20 h by DIC microscopy (Supplementary Figs. 22 and 24). A small number of thin fibrils were formed

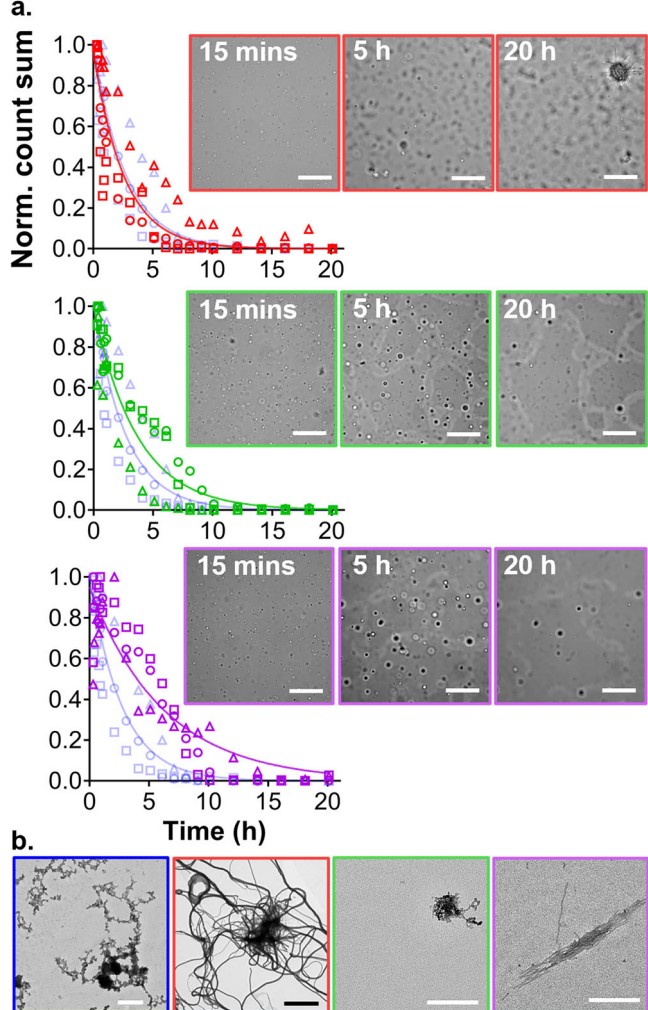

**Fig. 2 | Condensate and amyloid formation of truncated α-syn under phase separation conditions. a** Normalized total object count over time for 5-140 (red), 11-140 (green) and 19-140 (purple) α-syn. Three individual biological repeats are shown (circles, squares or triangles). Data are globally fitted using a one-phase decay model (solid lines). FL α-syn data (semi-transparent blue symbols and line) are shown for comparison. Overlayed are representative DIC images acquired *via* automation at the bottom of the well during incubation (scale bars represent 50 μm). **b** Representative TEM images of 5-140 (red), 11-140 (green) and 19-140 (purple) α-syn aliquots taken after 20 h incubation. Scale bars represent 400 nm, except for 5-140 α-syn whose scale bar is 4 μm.

at the end of the ThT intensity assay, consistent with delayed aggregation kinetics relative to phase-separated α-syn (Supplementary Figs. 15 and 25).

To obtain additional structural information, we compared the seeding ability of these fibrils to FL α-syn seeds (Supplementary Fig. 19). Unlike the FL seeds, which readily promoted aggregation, the truncated variants were markedly less effective, displaying reduced elongation capacity.

Together our data demonstrate that N-terminal truncation accelerates aggregation under conditions that induce phase separation. To understand if the faster maturation of the truncated variants was due to differences in phase partitioning, we quantified the dense- and dilute-phase concentrations and relative condensate volumes of FL and truncated α-syn. For FL α-syn, we estimated the total condensate volume to be ~ 0.2% and the protein concentration within condensates to be ~ 12 mM (Supplementary Fig. 26a–c), values in good agreement with existing literature[44,58], supporting the reliability of our assay. Importantly, 25 μM PLK did not absorb at 275 nm and so did not contribute to the absorbance spectra acquired in this assay (Supplementary Fig. 26d). Furthermore, no significant differences were detected between FL and truncated α-syn (Supplementary Fig. 26a–c),

indicating that the accelerated maturation of truncated α-syn is not due to altered phase partitioning. Instead, we propose that, with regards to 11-140 and 19-140 α-syn, faster maturation may be a consequence of the increased surface wettability of their condensates.

To further probe this mechanism, we asked whether differences in wettability and maturation could be explained by hydrophobicity, independently of charge. Thus, we purified and analyzed two additional N-terminally modified variants, each with the same Δcharge (−1) as 11-140 and 19-140 α-syn but with increased N-terminal hydrophobicity compared to FL α-syn (Supplementary Table 1 and Supplementary Fig. 27). The first was N-terminally acetylated FL α-syn (AcFL), a native modification known to influence membrane binding and aggregation[12,19,20,59–63]. The second was 14-140 α-syn, generated by deletion of residues 1–13; this variant corresponds to a fragment identified as a product of in vitro autoproteolytic α-syn cleavage[21]. The expectation was that if increased hydrophobicity reduced wettability, these variants would show less surface spreading and slower aggregation than FL α-syn. Instead, both variants showed extended condensate lifetimes in solution (Supplementary Fig. 28), and after 20 h, bottom-of-well imaging revealed enhanced wettability, consistent with 11-140 and 19-140 α-syn (Supplementary Fig. 29a). ThT assays further confirmed that AcFL and 14-140 α-syn aggregated faster than FL (Supplementary Fig. 29b–e). These results show that N-terminal hydrophobicity is not a key determinant of condensate wettability. Although all variants, bar 5-140 α-syn, had a Δcharge of −1, their enhanced wettability is better explained by greater exposure, due to weaker electrostatic interactions with the C-terminus, of N-terminal residues that can interact electrostatically with the negatively charged glass surface. Thus, it is the way charge is presented at the N-terminus, rather than hydrophobicity, that promotes surface wetting and accelerates amyloid formation.

## Reduced surface contact angle promotes α-syn amyloid formation

Our analysis of truncated variants suggests that surface wetting contributes to aggregation. To directly test this, we examined FL α-syn condensate wettability and maturation while varying the surface properties of the well. To do so, we incubated FL α-syn in 8-well slides with coverslips of varied hydrophobicity, *i.e.*, (i) a glass coverslip (hydrophilic surface) or (ii) a polymer coverslip (hydrophobic surface). As the surface wettability of a given protein condensate is largely determined by the balance between protein-protein interactions within the condensate versus condensate-surface interactions, we hypothesized that increasing surface hydrophobicity would disfavor condensate-surface interactions, thereby decreasing wettability. Additionally, (iii) a BioInert polymer coverslip, which is treated to render the surface inert to biomolecules, was also used as it is expected to further reduce condensate-surface interactions.

We prepared a solution of 60 μM α-syn and 25 μM PLK in phase separation buffer, supplemented with 20 μM ThT. Since direct fluorescence intensity measurements (*i.e.*, in a microplate reader) were not possible in the well slides, we imaged the samples using fluorescence microscopy and quantified the mean intensity of each image. Combining this analysis with DIC microscopy, we were able to simultaneously assess the morphology of surface-wetted condensates and the formation of ThT-positive species.

As expected, increasing well surface hydrophobicity reduced the wettability of α-syn condensates. This was evident from the increased contrast at the condensate border and the more spherical morphology of surface-wetted condensates (Fig. 4a, b). Additionally, the increase in condensate diameter after wetting was reduced on the more hydrophobic surface (Supplementary Fig. 30), further confirming lower wettability. Essentially, no surface wetting was observed on the BioInert coverslip (Fig. 4c and Supplementary Fig. 30).

Over time, the number of small ThT-positive aggregates increased for all surface types. However, aggregates were observed earlier on the glass and polymer coverslips, with notably more formed by the end of this assay (Fig. 4a–c). This suggests that increased wettability accelerates aggregation, a trend further supported by the mean fluorescence intensity of the images,

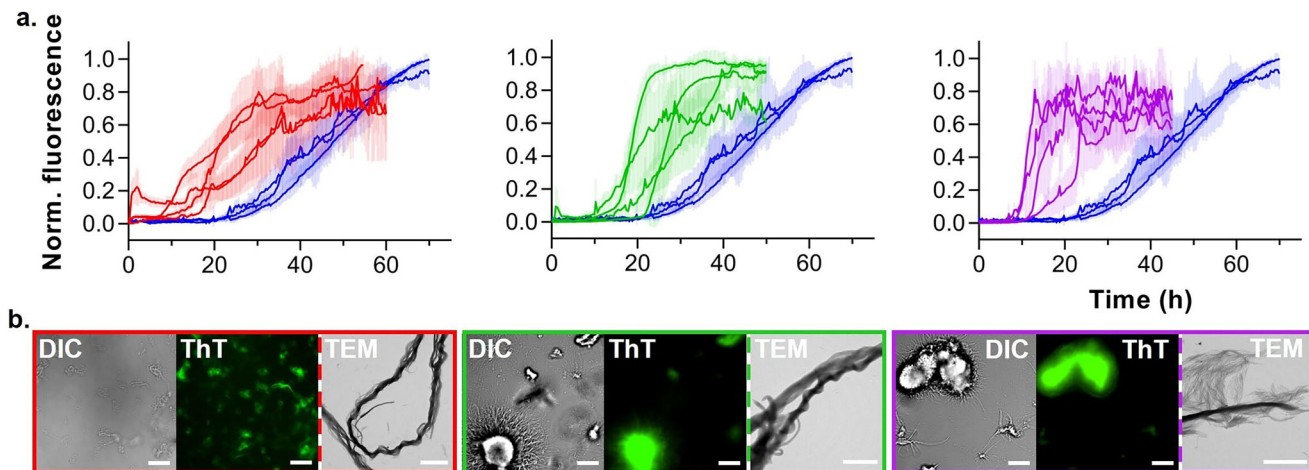

**Fig. 3 | Phase separation-induced amyloid formation of N-terminally truncated α-syn. a** Normalized ThT intensity of 5-140 (red), 11-140 (green) or 19-140 (purple) α-syn shown alongside the FL α-syn data (semi-transparent blue) as comparison. Each repeat is the mean of three technical replicates; error bars represent the standard deviation of the mean. **b** Representative DIC, ThT fluorescence, and TEM images taken at the bottom of the well at the endpoint of the assay. The fluorescence scale bars represent 25 μm; the TEM scale bars represent 3 μm.

**Fig. 4 | Wettability of phase-separated FL α-syn on different surfaces. a─c** Representative DIC (top) and fluorescent (bottom) images of 60 μM FL α-syn incubated with 25 μM PLK in phase separation buffer at 37 °C. Samples were incubated in 8-well slides with either a glass (**a**), polymer (**b**) or BioInert (**c**) coverslip. Images were acquired at selected time points at the bottom surface of the well. Scale bars represent 25 μm. ThT fluorescence intensity quantification of the microscopy images acquired during the screening assay of glass (**d**), polymer (**e**) or BioInert (**f**) coverslips. Each data point is the average fluorescent intensity across the whole image, 3 images were quantified per time point. The mean intensity per time point is indicated by a solid black line, error bars represent the standard deviation of the mean.

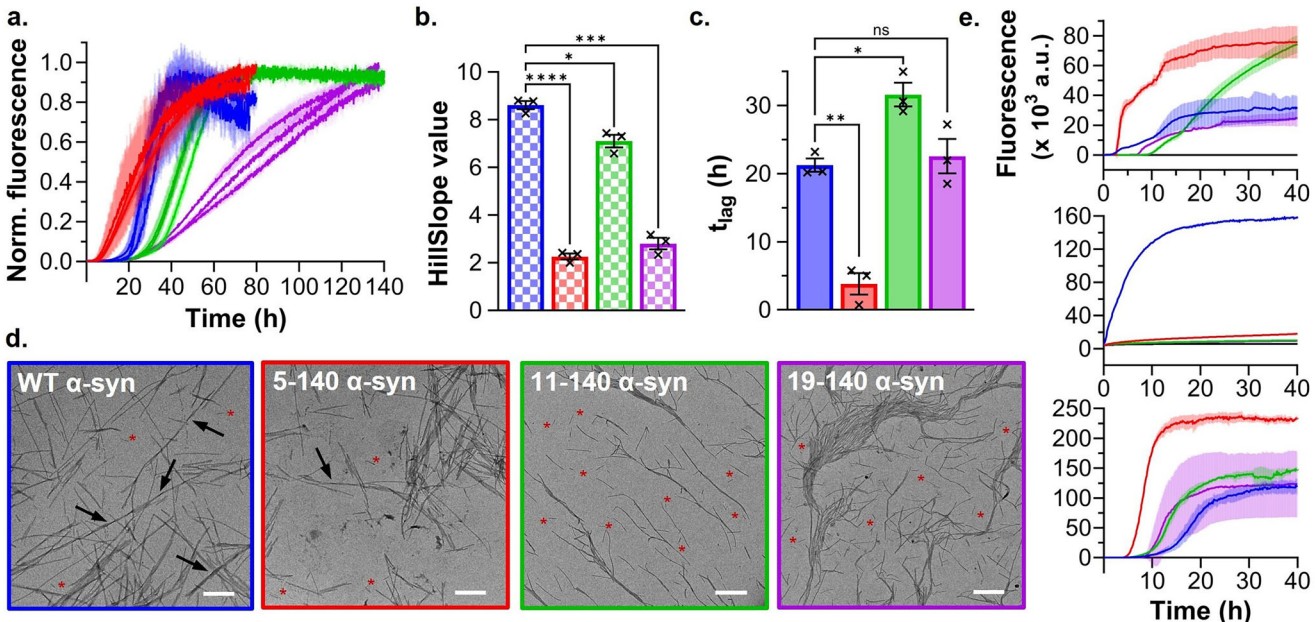

**Fig. 5 | The microscopic steps of α-syn amyloid aggregation. a** Dispersed solution aggregation of FL (blue), 5-140 (red), 11-140 (green) and 19-140 (purple) α-syn. Three individual biological repeats are shown. Each repeat is the mean of either two or three normalized technical replicates. The error bars represent the standard deviation of the mean. Bar plots representing the mean Hillslope (**b**) and $t_{lag}$ (**c**) values of FL (blue), 5-140 (red), 11-140 (green), and 19-140 (purple) α-syn. Error bars represent the standard error of the mean of $n = 3$ biological repeats, with individual data points represented by crosses. Statistical significance was determined using a Welch and Brown-Forsythe ANOVA with multiple comparisons against FL α-syn. p-values for 5-140, 11-140, and 19-140 α-syn were <0.0001, 0.0273, and 0.0002 for **b** and 0.0036, 0.0267, and 0.9235 for **c**, respectively. **d** Representative TEM images of the insoluble fraction after aggregation. Short fibrils are highlighted by a red asterisk and fibril twists are indicated by an arrow. Scale bars represent 500 nm. **e** Aggregation kinetics of FL (blue), 5-140 (red), 11-140 (green) and 19-140 (purple) α-syn under conditions promoting lipid-induced primary nucleation (top), elongation (middle), or secondary nucleation (bottom). A single representative biological repeat is shown. Each repeat is the mean of three technical replicates. The semi-transparent error bars represent the standard deviation of the mean.

which showed a marked increase relative to the buffer control only for the glass and polymer surfaces (Fig. 4d–f and Supplementary Fig. 31). These findings highlight the importance of surface wetting in α-syn condensate maturation, providing additional evidence that the increased wettability of 11-140 and 19-140 α-syn likely underpins their accelerated phase separation-induced aggregation, relative to the FL protein.

## The N-terminus regulates nucleation and propagation of α-syn fibrils

Our final aim was to analyze dispersed-solution amyloid aggregation to better understand why 5-140 α-syn shows distinct behavior compared to the other truncated variants. While aggregation within condensates likely follows different mechanisms from those under shaking or membrane-associated conditions, we used these conventional assays as benchmarks to test whether any of our PTMs consistently influence aggregation.

To monitor the kinetics of all three amyloid aggregation phases (*i.e.*, the lag, growth, and plateau phases) in a single assay, in the absence of phase separation, it was necessary to incubate α-syn with ThT under aggregation promoting conditions, *i.e.*, shaking in the presence of a glass bead at 37 °C (Fig. 5a and Supplementary Fig. 32). We then fitted the fluorescence data with a sigmoidal model to derive the time of the lag phase ($t_{lag}$) and the growth rate (Hillslope) of aggregation. We found that all N-terminally truncated α-syn variants aggregated more slowly overall in a dispersed solution than the FL protein, in agreement with previous reports on similar protein variants[34,36]. This behavior was mainly a consequence of significantly slower growth phases relative to FL α-syn, particularly 5-140 α-syn (Fig. 5a, b). In contrast, 5-140 α-syn had a significantly shorter lag phase than FL α-syn (~ 5.5-times faster $t_{lag}$), while those of 11-140 and 19-140 α-syn were comparable (~ 1.5- and 1.1-times slower $t_{lag}$, respectively) (Fig. 5a, c). Given the significant changes in the macroscopic phases of 5-140 α-syn aggregation, we decided to further characterize the aggregation

mechanism. Thus, we monitored dispersed solution aggregation kinetics at varied FL and 5-140 α-syn concentrations, such that the scaling exponent (γ), and hence the dominant aggregation mechanism, could be predicted using the AmyloFit software[64]. We found that while FL α-syn displayed fragmentation-dominated aggregation kinetics, *i.e.*, a linear double logarithmic plot with g ~ 0.5, 5-140 α-syn exhibited an increase in the contribution of secondary nucleation, leading to fragmentation and secondary nucleation-dominated aggregation kinetics (Supplementary Fig. 33)[64].

Subsequently, we quantified the percentage of monomeric protein converted into aggregates during the assay. We found similar amounts of soluble protein remained at the end of aggregation for all variants (~ 10–20% of the initial monomer), indicating a similar conversion of monomers into aggregates (Supplementary Fig. 34). Finally, we characterized the morphology and size of the aggregates by performing TEM on the insoluble protein fractions isolated after the dispersed solution aggregation assay (Fig. 5d and Supplementary Figs. 35 and 36). All variants formed straight fibrils; however, only FL and 5-140 α-syn also formed twisted fibrils.

Quantitative analysis of the TEM images highlighted that while FL and 5-140 α-syn fibrils were similar in length, 11-140 and 19-140 α-syn fibrils were typically shorter (Fig. 5d and Supplementary Figs. 35 and 36). Hence, increasing N-terminal truncation appears to reduce the structural stability of α-syn fibrils formed in a dispersed solution. This result is consistent with existing evidence that truncation of the first 6 N-terminal residues reduces fibril stability against mechanical agitation and protease activity[34]. As the amyloids formed during our dispersed solution aggregation assay experience significant mechanical agitation due to shaking, our data indicate that deleting residues 5 and 6 destabilizes α-syn amyloids against mechanical stress-induced fragmentation.

5-140 and FL α-syn fibrils also shared a comparable median width, while both 11-140 and 19-140 α-syn amyloids were significantly thinner (Fig. 5d and Supplementary Figs. 35 and 36). Thus, 11-140 and 19-140 α-syn

fibrils were likely at an earlier maturation stage, *i.e.*, protofilaments and/or protofibrils. This theory is supported by the high degree of fibrillar clumping observed for 11-140 and 19-140 α-syn, where many thin fibrils clustered together into large bundles (Fig. 5d and Supplementary Fig. 35). Such bundles may result from non-specific interactions between exposed hydrophobic regions on the protofilaments/protofibrils. Our findings are consistent with the inability of 7-140 α-syn to form thick fibrils, as we have previously reported[34], indicating deletion of residues 5 and 6 arrests dispersed solution aggregation at an earlier maturation stage.

Together, our data demonstrate that N-terminal truncation has a widespread effect on the dispersed solution amyloid aggregation of α-syn.

### The N-terminus regulates interaction of α-syn monomers with lipids and fibrils

To further our understanding of the role played by the N-terminus in the lag and growth phases of α-syn dispersed-solution aggregation, we examined the microscopic aggregation mechanisms: primary nucleation, secondary nucleation, and fibril elongation, for each variant. To ensure that all pre-formed fibrils used as seeds were structurally mature and homogeneous, we used fibrils made of FL α-syn. Indeed, aggregates of N-terminally truncated α-syn formed in a dispersed solution were morphologically distinct from FL α-syn fibrils, suggesting arrest at different maturation stages (Fig. 5d). The size distribution of all pre-formed fibril seeds and lipid vesicles was confirmed by DLS (Supplementary Fig. 37).

All truncated variants showed markedly reduced elongation compared to FL α-syn (Fig. 5e). Consistent with previous evidence[36], our data indicate that the first 4 residues of α-syn monomers are critical for their interaction with fibril ends, explaining the delayed growth phase observed during the dispersed solution aggregation assay. Alongside secondary nucleation, we also assessed the ability of each variant to undergo heterogeneous primary nucleation in the presence of 100 nm DMPS lipid vesicles. 11-140 and 19-140 α-syn displayed comparable aggregation kinetics, with inhibited lipid-induced primary nucleation and enhanced secondary nucleation, while both nucleation mechanisms were significantly accelerated for 5-140 α-syn (Fig. 5e). Given 5-140 α-syn lacks Asp2, decreased electrostatic repulsion towards the negatively charged DMPS lipid vesicles may promote primary nucleation. In contrast, the increased negative charge and absence of key lipid-binding residues in both 11-140 and 19-140 α-syn likely inhibit this nucleation[28]. Under secondary nucleation conditions, where the pH approaches α-syn's isoelectric point and so electrostatic effects are minimized, the decreased hydrophobicity of all truncated variants may reduce hydrophilic-hydrophobic repulsion between α-syn monomers and the hydrophilic C-terminal fuzzy coat on fibrils, thus promoting secondary nucleation.

## Discussion

Here, we used a set of pathologically relevant N-terminal truncations to provide a detailed mechanistic description of α-syn phase separation and condensate maturation. To do so, we developed a strategy combining the use of microscopy and a microplate reader to monitor and quantify condensate formation, surface wetting, and amyloid aggregation. To model phase separation in vitro, we used PEG-8000 and PLK, well-established agents that mimic macromolecular crowding and polycationic interactions, respectively. While this in vitro system is a simplified representation of the complex crowding and protein–protein interactions that α-syn experiences in vivo, it still provides valuable insights into the fundamental mechanisms of α-syn phase separation. In particular, our findings highlight the importance of surface wetting in modulating α-syn behavior, which is in line with recent studies demonstrating that biological surfaces can influence phase separation[55,56].

We found that N-terminal truncation had little effect on the initial formation of α-syn condensates but strongly influenced their maturation into amyloids (Fig. 3). Two effects were observed: (1) truncated α-syn condensates persisted longer in the bulk and (2) they exhibited increased wettability on the glass surface. Slower sedimentation delays coalescence between suspended

condensates, thereby prolonging the time the system remains in a high surface-to-volume state. Increased wettability further promotes spreading on the surface, effectively enlarging the condensate–surface interface. Together, these effects mean that truncated variants maintain a higher effective surface-to-volume ratio than FL α-syn. Further evidence was obtained by modulating condensate-surface interactions, which demonstrated that increased wettability accelerated FL α-syn aggregation (Fig. 4).

To further investigate why the truncated variants showed enhanced wettability, we analyzed two extra N-terminally modified variants, 14-140 and AcFL α-syn. Both share the same Δcharge (−1) and higher N-terminal hydrophobicity than FL α-syn, with 14-140 being the most hydrophobic construct tested. We expected that if increased hydrophobicity reduced surface wetting, these variants would aggregate slower than the FL protein. Yet, both displayed prolonged condensate lifetimes in solution, enhanced wettability, and faster aggregation (Supplementary Figs. 28 and 29). These findings demonstrate that α-syn condensate wettability is not reduced by increased N-terminal hydrophobicity.

Instead, we propose that, for AcFL α-syn, N-terminal acetylation neutralizes the free amine and stabilizes a helical conformation that promotes interaction with negatively charged surfaces[59,61], consistent with its increased wettability. In the case of truncated α-syn, removal of charged and amphipathic residues may alter the extent of N-terminal exposure, which may similarly enhance surface interactions and accelerate condensate maturation.

The notable exception to this trend was the pathological variant 5-140 α-syn, where aggregate formation was observed within suspended condensates (Fig. 3 and Supplementary Fig. 22). AmyloFit analysis of 5-140 α-syn dispersed solution aggregation kinetics predicted an increased contribution of secondary nucleation (Supplementary Fig. 33). This was confirmed by investigation into the microscopic aggregation mechanisms, which revealed significantly enhanced interface-catalyzed nucleation of 5-140 α-syn on both lipid vesicles and pre-formed fibrils (Fig. 5). Thus, 5-140 α-syn may aggregate faster within condensates because its nucleation at the condensate-bulk solution interface is also accelerated.

Conversely, our dispersed-solution aggregation analysis indicates that certain N-terminal truncations (*i.e.*, 1–10 and 1–18) yield protofilament-like assemblies, suggesting that these modifications impair both aggregation kinetics and fibril maturation. However, under phase separation conditions, all truncated variants aggregated more rapidly, consistent with enhanced wettability facilitating nucleation. These observations highlight that, although increased nucleation propensity may accelerate aggregation under phase separation conditions, fibril formation and phase separation are governed by distinct mechanisms. Nonetheless, our data suggest that N-terminal truncations bias aggregation toward surface-dependent nucleation pathways.

Overall, our data show that α-syn condensate maturation is influenced by wettability. This effect is likely mediated by changes in condensate surface-to-volume ratio, which modulate aggregation at the condensate–bulk solution interface. This is consistent with previous reports that α-syn can aggregate at the surface of pre-formed coacervates composed of inert biomolecules[45]. In the context of neurodegeneration, changing α-syn condensate wettability within cells through therapeutic intervention (*e.g.*, through targeting the N-terminus) could be a viable avenue towards mitigating α-syn toxicity. Our findings also indicate that α-syn phase separation and amyloid aggregation are linked, yet able to proceed independently. In fact, the same protein modification may have distinct effects on each pathway. This suggests that inhibitors of one process will not necessarily affect the other, a possibility that should be taken into consideration for future drug discovery. Our designed techniques establish a robust foundation for future investigations into the microscopic mechanisms mediating α-syn phase separation and amyloid aggregation.

## Methods

### Differential interference contrast and fluorescence microscopy

To assess the phase separation behavior of α-syn, the protein was incubated with PEG-8000 and PLK, conditions previously shown to induce α-syn

phase separation[42]. PEG-8000 was dissolved in PBS, pH 7.4 to a concentration of 20% w/v. PLK hydrochloride with a molecular weight of 15,000–30,000 g/mol was dissolved in 10 mM HEPES, 100 mM NaCl, pH 7.4 to a concentration of 3 mM (determined using an approximate molecular weight of 22,500 g/mol). Phase separation was induced by incubating monomeric protein (0, 20, 40, and 60 μM) with PLK (0, 25, 50 and 100 μM) in phase separation buffer (PBS, pH 7.4, 10% PEG-8000, 0.02% NaN$_3$). 5 minutes after PLK addition, 100 μl of sample (single replicate per assay) was loaded into a 96-well full-area glass-bottom plate (Sensoplate, Greiner Bio One, Austria). The plate was covered with a clear, microscopy compatible self-adhesive film (ibiSeal, Ibidi, Germany). The film was secured onto the plate using strips of adhesive aluminum foil along each edge. The samples were incubated at 37 °C within the chamber of a Nikon ECLIPSE Ti2-E microscope (Nikon, Japan) and automated DIC microscopy imaging in solution was carried out 15 min from PLK addition using a 40x air objective. A z-stack (3-step, 200 μm step size) within the bulk solution was acquired per well.

Condensate maturation in samples containing monomeric α-syn (0 or 60 μM) and PLK (0 or 25 μM) was monitored at 37 °C using the above automated solution imaging at pre-determined time points over a 20 h incubation period. Additionally, automated imaging was performed on the bottom surface of the well at each time point. Endpoint (bottom of well and in solution) DIC images were manually captured after the 20 h time point. When stated, 20 μM ThT was included to monitor amyloid fibril formation, and fluorescence images were taken alongside DIC images using excitation 405 nm and emission 515/30 nm filters. Unless otherwise stated, a 40x air objective was used.

Both the 15 min and time course DIC solution images underwent processing, which involved estimating the number of objects per image and the area of every object in a given image. This analysis was conducted using GA3 software integrated into the NIS-Elements software (Nikon, Japan). A detailed GA3 recipe is provided in Supplementary Fig. 3. In brief, images were first subjected to denoising to reduce shot noise. Objects were identified within the images by defining a gain thresholding range. To enhance accuracy, border objects and objects exceeding 50 μm² in area (larger than the maximum condensate area determined through manual inspection) were excluded. Furthermore, the objective used has a numerical aperture of 0.6. Approximating the wavelength of light as 550 nm, the spatial resolution of this objective is 0.556 μm. Thus, objects with an area of 0.243 μm² or less were excluded. The remaining object area values and the total number of objects per image, across all z positions, per well, and time point were consolidated into a single data table.

Phase diagrams were generated using the normalized total number of objects counted per well, summed over all z positions, 15 min from PLK addition. To examine the time course experiments, the area distribution of each sample, compiled over all z positions, at each time point, was plotted. The mean area at each time point was determined, plotted against time, and normalized. Mean area values for time points where the number of condensates is ≤ 33% of the initial number of condensates were excluded, as the low sample size leads to unreliable mean values. Three separate biological repeats were globally fitted with a one-phase association curve. The half-time ($t_{50}$) of object growth was derived. Next, the total object count at each time point, compiled over all z positions, was plotted against time and normalized. Three separate biological repeats were globally fitted with a one-phase decay curve and the half-time ($t_{50}$) of count decay was derived. All data were plotted using GraphPad Prism version 10.0.3.

FL α-syn localization was monitored using a 1:100 ratio of rhodamine labeled:unlabeled FL α-syn. Fluorescence images were taken alongside DIC images using excitation 525 nm and emission 641/75 nm filters. A single sample was prepared at a time, loaded into a 96-well full-area glass-bottom plate, and imaged manually using a 40x air objective.

### Aggregation assay under phase separation conditions
Fibril formation was monitored by incubation of monomeric α-syn (0 and 60 μM) with PLK (0 and 25 μM) and 20 μM ThT in phase separation buffer.

100 μl of sample (3 replicates) was loaded into a 96 well full-area glass-bottom plate, sealed with aluminum foil, and incubated at 37 °C under quiescent conditions for ~ 70 h in a CLARIOstar Plus microplate reader. Fluorescent intensity measurements were taken at the center of the well, excitation 440-10 nm, dichroic 460 nm and emission 480-10 nm filters, 2 gains and 30 flashes per well, every 20 min. All sample spectra were background corrected by subtracting the spectrum of 20 μM ThT in phase separation buffer, with data extracted from the same assay background corrected using the same buffer spectrum. The data were plotted using GraphPad Prism version 10.0.3.

### Phase separation in the presence of different surfaces
Surface wetting and aggregate formation were monitored by incubation of 0 or 60 μM FL α-syn with 25 μM PLK and 20 μM ThT in phase separation buffer. 200 μl of sample was loaded into μ-slide 8-well$^{high}$ (Ibidi, Germany) with either a glass coverslip, a polymer coverslip, or a BioInert polymer coverslip. Samples were incubated at 37 °C for 2 d and the bottom of the well imaged at selected time points by DIC and ThT fluorescence microscopy (excitation 405 nm, emission 515/30 nm filters) using a 60x oil objective. The mean fluorescence intensity of each image was measured and plotted using GraphPad Prism version 10.0.3.

### Reporting summary
Further information on research design is available in the Nature Portfolio Reporting Summary linked to this article.

### Data availability
The source data corresponding to any statistical analysis and all full-length, unprocessed electrophoresis gel and dot-blot images, and microscopy images used for quantitative analysis, that support the findings of this study are available in the Zenodo repository, https://doi.org/10.5281/zenodo.13335999.

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

## Acknowledgements

We thank the UK Research and Innovation (Future Leaders Fellowship MR/S033947/1 and Fellowship renewal MR/Y003616/1 to F.A.A.), the Engineering and Physical Sciences Research Council (grant EP/S023518/1), Alzheimer's Society, UK (grant 511), Alzheimer's Research UK (ARUK-PG2019B-020), and the Department of Chemistry of Imperial College London (PhD studentship to R.J.T.). We also thank Prof. Dr. Paolo Arosio and Prof. Annalisa Pastore for helpful discussions and feedback on the manuscript. Furthermore, we thank Lisa Haigh, Malgorzata Puchnarewicz, and Adriana Lobosco in the Chemistry Mass Spectrometry Facilities for assistance with ESI-MS, and the Electron Microscopy Centre facilities at The Center of Structural Biology for assistance with TEM. Figure 1a includes a darkened version of the multiwell-plate-3d icon by Servier, https://smart.servier.com/ is licensed under CC-BY 3.0 Unported https://creativecommons.org/licenses/by/3.0/.

## Author contributions

R.J.T., D.M.V. and S.C.A. performed all experiments. R.J.T. and F.A.A. analyzed the data. R.J.T. and F.A.A. conceptualized the work. R.J.T., D.M.V., S.C.A., M.S., and F.A.A. discussed the data and provided feedback on the manuscript.

## Competing interests

The authors declare no competing interests.
