## [Transparent Peer Review file · Communications Chemistry]

Surface Wetting Is a Key Determinant of α -Synuclein Condensate Maturation

Corresponding Author: Dr Francesco Aprile

Version 0:

Reviewer comments:

Reviewer #1

(Remarks to the Author)

The authors report the study of α -synuclein condensates maturation as a function of surface wetting. The article is scientifically sound and deserve publication in Communications Chemistry. However, several key points need to be addressed before publication:

- In the abstract the authors write: our results “indicate that condensate wetting on cellular surfaces, such as synaptic vesicles, may drive toxic aggregate formation during neurodegeneration”. This conclusion is not supported by the data presented here and showing wetting on very different surfaces than lipid membranes.
- The authors present in Figure 1c the objects area distribution and show points smaller than 0.1 μm^2 . Is that possible with the resolution of their confocal microscope? Which is the spatial resolution of the microscope used?
- In SI Figure S4 both α -syn and PLK are given in μM to make the phase diagrams of LLPS. Yet, from line 97 and following lines in the text the authors write that FL α -syn form condensates at 40 or 60 mM with 25 mM PLK. Should all concentrations be in μM or only α -synuclein concentration? Although it might be just a typo it is a critical comment to address.
- Figure 1f does not show twisted fibrils, the authors should present quantitative data of the twist of the fibrils.
- The authors comment “our data demonstrate that N-terminal truncation has a widespread effect on the dispersed solution amyloid aggregation of α -syn”. In Figure S31 the authors comment on structural differences between amyloid fibrils of differently truncated α -synuclein protein. The data showed in the figures may not represent statistically significant differences. The authors are encouraged to represent the data as box plots and compared with a statistical test the different samples.
The authors should also account the relative error that is larger on the width measurements than for the height by TEM. How are they sure the measured width is not affected by drying effects? Considered that TEM makes difficult to distinguish single fibrils from bundles? How is the structural analysis performed?
- The authors say that all variants formed rod-like fibrils, but a more quantitative analysis of the TEM images and complementary methods such as AFM should be used to support this conclusion.
- In the conclusions, the authors place their results in the context of LLPS in amyloid formation in neurodegeneration and drug discovery approaches. How the use of PEG and PLK would relate to the strength of these conclusions? The reviewer would suggest to highlight as this system leveraging artificial crowding agents might differ from α -synuclein in physiological conditions.

Reviewer #2

(Remarks to the Author)

Reviewer comments:

The manuscript by Thrush et al. aims to tackle a very important challenge in the field of disease related biomolecular

condensates—to study the effect of surface wetting on its propensity to form amyloid fibrils. They have studied how a-Syn, the protein involved in Parkinson's disease pathogenesis aggregates within model coacervates when they sediment on surfaces, and compared how the N-terminal truncation alters this behavior. The work is timely, and the data presented are somewhat convincing. However, the claims by the authors are not very robust due to a lack of additional important information. I have three main concerns in this manuscript which I indicate below.

Major concerns:

The authors claim to have established a quantitative framework to understand how condensate wettability affects amyloid fibril formation within these assemblies. However, apart from measuring the condensate number and fluorescence intensity, they do not quantify any other physicochemical parameters. This is a very serious concern because only these two parameters alone may not be enough to provide a quantitative understanding of amyloid aggregation on condensate surfaces—especially in a comparative setting (with mutants).

1. The authors do not quantify the dense/dilute phase concentrations after phase separation, neither do they comment on the volume fractions of the two phases for WT and different mutants. Without appreciating the differences in their dense/dilute phase concentrations, and normalizing that for WT and mutants, the exclusive effect of condensate surface wetting on amyloid aggregation cannot be truly quantified. This is because if the concentration of the dense phase is even one order of magnitude different in a mutant compared to WT (at a given solution condition), the kinetics of amyloid aggregation within that mutant condensate will be greatly different compared to WT—even without the effects arising from surface wetting.
2. It is very confusing to understand from reading the manuscript whether the accelerated aggregation is due to the increased surface to volume ratio of the mutant condensates (due to altered fusion/ripening mechanisms) or due to the differential contact with the new surface upon sedimentation. What exactly promote amyloid nucleation? Is it the new surface upon sedimentation or is it the surface to volume ratio of the condensates? Let's say there are 100 small condensates versus 1 large condensate—can we expect more aggregation in 100 smaller condensates due to a higher available surface area? Again, this effect (size distribution and volume fraction of the dense phase) should have been considered before studying the effect of sedimentation induced amyloid aggregation across mutants.
3. The authors compare the aggregation effects of the WT and mutant proteins under two other, more conventional settings—using shaking/fragmentation and on membrane surfaces—Why? The condensate mediated pathway, the amyloid aggregation kinetics, the dominant fibril polymorph formed—may be completely different under the phase separating setting compared to the other two. The differences in the kinetic parameters of the mutants under these two conditions may have nothing to do with mechanisms via the condensate pathway.

Minor concerns:

1. Due to the ever-increasing complexity of biomolecular phase separation, it is now more relevant to call it just phase separation (and not LLPS). Especially because we now know that they are often coupled with other phase transition mechanisms such as percolation and micellization (Kar et al. PNAS, 2022).
2. Line 58: "Together, these findings indicate that electrostatic interactions mediate α -syn condensate formation, suggesting a key role for the terminal domains. Furthermore, b-syn, which lacks the hydrophobic eight residue stretch located in the non-amyloid-b component (NAC) domain of α -syn, does not phase separate or aggregate, indicating that this domain is critical for LLPS". These statements are confusing. Full length a-Syn phase separates at higher Debye screening—making them electrostatically unfavorable (Ray et al. Nat. Chem. 2023; BioRxiv. 2024). The second statement contradicts the first one—the first statement says it is mediated via the terminal domains via electrostatics and the second statement highlights the importance of the NAC domain. I don't understand what do the authors try to communicate here.
3. The protein concentrations are written in mM (millimolar) in the main manuscript while in μ M (micromolar in the SI).
4. The PLK-aSyn fibrils could be mixed fibrils. The authors should highlight this more prominently in the main manuscript.
5. Line 176: "...supporting our earlier hypothesis that condensate growth plateaus when their number falls below a critical threshold". This is not your hypothesis. This is classical condensate growth kinetics via fusion/ripening.
6. The manuscript has some awkward phrasing that breaks the flow of reading.

Suggestions to the authors:

Overall, my suggestion would be to calculate the volume fractions and the concentrations of the dense and the dilute phase of condensates formed by the WT and mutant proteins and normalize these parameters to study the effect of surface wetting exclusively. Also, the authors are encouraged to quantify the concentration of elongation competent amyloid fibrils formed in each case with a classical seeding assay (and not just comment on their number under TEM).

Reviewer #3

(Remarks to the Author)

This study aims at providing a platform that monitors and quantifies condensate formation, surface wetting, and amyloid aggregation for different alpha-synuclein variants. I believe that trying to systematically establish the relationship amongst these process, if any, is an important question to address. The approach combining 96-well plates with DIC and fluorescence microscopy and Thioflavin-T measurements is technically sound. The authors focus on 4 synuclein variants, WT and 3 which are truncated at the N-terminus. The overall conclusion is that larger truncations, which substantially reduce hydrophobicity, slow down condensate sedimentation, make the condensates more wettable and speeding up amyloid formation. The decoupling between the process of condensate sedimentation, overall slower for these variants, and their ability to aggregate into amyloids is an important finding. The study further highlights the role of the surface to volume ratio in modulating amyloid formation due to the catalytic role of the interface at the condensate surface.

I think this study should be published but I see some limitations which should be addressed to ensure the robustness of the message:

1. The study characterises 4 variants (including WT). Given that one of them (5-140) has a different behaviour from the

others, it is hard to draw very robust general conclusions. The authors suggest that changes in hydrophobicity are responsible for the different sedimentation, gettability and aggregation pattern. Can they design a couple of other variants (the more the better), with similar hydrophobicity profiles, and assess their behaviour to test their overall hypothesis? This would also further validate the high-throughput feature of their experimental set-up.

2. In line with the above, occasionally one single biological replicate is presented (Figure 3). Is this due to extreme variability across biological replicates? Otherwise, the set-up seems ideal to have multiple measurements for one same protein.

3. Data in Figure 5 adds an interesting layer as the author show and discuss the different effects that truncations have on the kinetics of aggregation in the soluble phase, as well as in the stability of the final fibrils. I think these observations are not trivial and would appreciate a sentence on the relationship between kinetics and stability (of condensates and of fibrils) in the discussion.

4. Figure 5, bottom right - y-axis label is missing.

Version 1:

Reviewer comments:

Reviewer #1

(Remarks to the Author)

The authors have satisfactorily addressed my comments, the manuscript is suitable for publication in Communications Chemistry.

Reviewer #2

(Remarks to the Author)

The authors have done a good job in addressing all my concerns. I have no further comments.

Reviewer #3

(Remarks to the Author)

The authors did a good job at addressing mine and the other reviewers' points.

made.

We would like to thank the reviewers for their valuable feedback. In response, we have carried out several new experiments, including additional replicates, concentration measurements in the dense and dilute phases, characterization of the seeding properties of fibrils formed under phase separation conditions, and analysis of additional protein variants. Below, we provide a point-by-point response (blue), which we hope addresses all concerns (black).

Reviewer #1

The authors report the study of α -synuclein condensates maturation as a function of surface wetting. The article is scientifically sound and deserve publication in Communications Chemistry. However, several key points need to be addressed before publication:

1. In the abstract the authors write: our results “indicate that condensate wetting on cellular surfaces, such as synaptic vesicles, may drive toxic aggregate formation during neurodegeneration”. This conclusion is not supported by the data presented here and showing wetting on very different surfaces than lipid membranes.

We thank the reviewer for their supportive comments on our manuscript. We agree with the reviewer’s observation and would like to clarify that our intention was to highlight the potential biological relevance of interfaces, as recent studies have shown that biological surfaces can influence liquid-liquid phase separation. We did not intend to imply that we directly studied wetting in the presence of cellular surfaces. To avoid any ambiguity, we have rephrased the abstract to improve the clarity of our conclusion (page 1 of the main text), as follows:

Our results reveal distinct mechanistic roles for α -synuclein N-terminal residues and highlight that condensate wettability and the condensate-bulk solution interface play an important role in modulating toxic aggregate formation.

2. The authors present in Figure 1c the objects area distribution and show points smaller than 0.1 μm^2 . Is that possible with the resolution of their confocal microscope? Which is the spatial resolution of the microscope used?

The 40 \times objective used has a numerical aperture of 0.6. Approximating the wavelength of light as 550 nm, the spatial resolution is $\sim 0.56 \mu\text{m}$. Accordingly, objects with a radius $\leq 0.28 \mu\text{m}$ (area $\leq 0.24 \mu\text{m}^2$) were excluded from the analysis.

3. In SI Figure S4 both α -syn and PLK are given in μM to make the phase diagrams of LLPS. Yet, from line 97 and following lines in the text the authors write that FL α -syn form condensates at 40 or 60 mM with 25mM PLK. Should all concentrations be in μM or only α -synuclein concentration? Although it might be just a typo it is a critical comment to address.

We thank the reviewer for pointing out this typo. The concentrations have been corrected, and both α -syn and PLK are now consistently reported in μM throughout the text.

4. Figure 1f does not show twisted fibrils, the authors should present quantitative data of the twist of the fibrils.

We thank the reviewer for this observation. We have revised the text to use more accurate language, clarifying that the fibrils are spiraled rather than twisted. We do not provide quantitative measurements of fibril twist here but will consider this in future work.

5. The authors comment “our data demonstrate that N-terminal truncation has a widespread effect

on the dispersed solution amyloid aggregation of α -syn". In Figure S31 the authors comment on structural differences between amyloid fibrils of differently truncated α -synuclein protein. The data showed in the figures may not represent statistically significant differences.

A. The authors are encouraged to represent the data as box plots and compared with a statistical test the different samples.

We thank the reviewer for the suggestion. The data are now presented as box plots, and we have included statistical analysis using ANOVA with multiple comparisons against the control (FL α -syn).

B. The authors should also account the relative error that is larger on the width measurements than for the height by TEM.

We thank the reviewer for this comment. We have clarified this limitation in the text. Importantly, we also note that the comparative trends across variants remain valid, since all fibrils were imaged under identical conditions and are equally affected by the limited lateral resolution of negative-stain TEM (page 3 of the SI):

Because fibrils were identified by the presence of stain along their edges and given the limited lateral resolution of negative-stain TEM, width measurements were associated with a higher relative error than length measurements. To account for this, we analyzed ≥ 292 fibrils per condition, with statistical significance assessed using ANOVA.

C. How are they sure the measured width is not affected by drying effects?

We thank the reviewer for this comment, drying effects in TEM samples cannot be fully excluded, however as our aim was to compare fibril morphologies between variants, this approach remains valid. We have stressed this point in the revised text. We have revised the text as follows (page 3 of the SI):

Sample application, washing and staining steps were performed in rapid succession to ensure that the samples did not dry out between steps as dehydration has been shown to destabilize amyloid fibrils, leading to changes in their appearance if rehydrated.^{1,2}

D. Considered that TEM makes difficult to distinguish single fibrils from bundles? How is the structural analysis performed?

We have clarified this in the revised manuscript to avoid ambiguity (page 3 of the SI):

Only distinct amyloid fibrils that were clearly distinguishable as individual structures, not fibril bundles or clusters, that were wholly within the image were measured and a similar number of images at similar magnifications were quantified across variants.

6. The authors say that all variants formed rod-like fibrils, but a more quantitative analysis of the TEM images and complementary methods such as AFM should be used to support this conclusion.

We have revised the text to use more accurate language, clarifying that we describe the aggregates qualitatively as straight fibrils rather than making a quantitative claim about their morphology. We fully agree with the reviewer that a more quantitative analysis of fibril morphology,

including detailed TEM evaluation and complementary methods such as AFM, would further strengthen this conclusion. Such an analysis, however, is beyond the scope of the present work and will be included in a follow-up study.

7. In the conclusions, the authors place their results in the context of phase separation in amyloid formation in neurodegeneration and drug discovery approaches. How the use of PEG and PLK would relate to the strength of these conclusions? The reviewer would suggest to highlight as this system leveraging artificial crowding agents might differ from α -synuclein in physiological conditions.

Following the reviewer's suggestion, we have revised the Conclusions section to better emphasize the potential differences between our *in vitro* system (page 9 of the main text), as follows:

To model phase separation in vitro, we used PEG-8000 and PLK, well-established agents that mimic macromolecular crowding and polycationic interactions, respectively. While this in vitro system is a simplified representation of the complex crowding and protein–protein interactions that α -syn experiences in vivo, it still provides valuable insights into the fundamental mechanisms of α -syn phase separation. In particular, our findings highlight the importance of surface wetting in modulating α -syn behavior, which is in line with recent studies demonstrating that biological surfaces can influence phase separation.^{3,4}

Reviewer #2

The manuscript by Thrush et al. aims to tackle a very important challenge in the field of disease related biomolecular condensates—to study the effect of surface wetting on its propensity to form amyloid fibrils. They have studied how α -Syn, the protein involved in Parkinson's disease pathogenesis aggregates within model coacervates when they sediment on surfaces, and compared how the N-terminal truncation alters this behavior. The work is timely, and the data presented are somewhat convincing. However, the claims by the authors are not very robust due to a lack of additional important information. I have three main concerns in this manuscript which I indicate below.

Major concerns:

The authors claim to have established a quantitative framework to understand how condensate wettability affects amyloid fibril formation within these assemblies. However, apart from measuring the condensate number and fluorescence intensity, they do not quantify any other physicochemical parameters. This is a very serious concern because only these two parameters alone may not be enough to provide a quantitative understanding of amyloid aggregation on condensate surfaces—especially in a comparative setting (with mutants).

We thank the reviewer for raising this important point. To address it, we have performed new experiments, as outlined below, to quantify additional physicochemical parameters of the various protein variants.

1. The authors do not quantify the dense/dilute phase concentrations after phase separation, neither do they comment on the volume fractions of the two phases for WT and different mutants. Without appreciating the differences in their dense/dilute phase concentrations, and normalizing that for WT and mutants, the exclusive effect of condensate surface wetting on amyloid aggregation cannot be truly quantified. This is because if the concentration of the dense phase is even one order of magnitude different in a mutant compared to WT (at a given solution condition),

the kinetics of amyloid aggregation within that mutant condensate will be greatly different compared to WT—even without the effects arising from surface wetting.

We have performed additional experiments to estimate the dense- and dilute-phase concentrations, as well as the relative volumes, for FL and N-terminally truncated α -syn. The two phases were separated by centrifugation, with speed and duration determined using a previously described equation.⁵ Protein concentrations in the dilute phase were then measured *via* absorbance. To estimate the condensate volume fraction, fluorescence microscopy images acquired after 15 min incubation were analyzed by dividing the total condensate area per image by the total image area, and the mean % area value was calculated from three z-stack images per biological replicate. No significant differences were observed between the variants. These results are now included in the revised manuscript (Supplementary Fig. 26).

Figure S26. N-terminal truncation does not significantly alter phase partitioning.

a, Volume fraction of FL (blue), 5-140 (red), 11-140 (green) and 19-140 (purple) α -syn condensates. b/c, Concentration of FL (blue), 5-140 (red), 11-140 (green) and 19-140 (purple) α -syn in the dilute-phase (b) and the dense-phase (c). d. Supernatant absorbance after centrifugation of 25 μ M PLK in phase separation buffer. Three biological replicates are shown, the spectra have been buffer-subtracted and baseline corrected nm.

2. It is very confusing to understand from reading the manuscript whether the accelerated aggregation is due to the increased surface to volume ratio of the mutant condensates (due to altered fusion/ripening mechanisms) or due to the differential contact with the new surface upon sedimentation. What exactly promote amyloid nucleation? Is it the new surface upon sedimentation or is it the surface to volume ratio of the condensates? Let's say there are 100 small condensates versus 1 large condensate—can we expect more aggregation in 100 smaller

condensates due to a higher available surface area? Again, this effect (size distribution and volume fraction of the dense phase) should have been considered before studying the effect of sedimentation induced amyloid aggregation across mutants.

We thank the reviewer for this comment, as we did not detect significant differences in condensate volume or concentration across the variants, we have clarified our conclusion as follows (page 9 of the main text):

Slower sedimentation delays coalescence between suspended condensates, thereby prolonging the time the system remains in a high surface-to-volume state. Increased wettability further promotes spreading at the surface, effectively enlarging the condensate–surface interface. Together, these effects mean that truncated variants maintain a higher effective surface-to-volume ratio than FL α -syn, providing a mechanistic explanation for their accelerated maturation into amyloids.

3. The authors compare the aggregation effects of the WT and mutant proteins under two other, more conventional settings—using shaking/fragmentation and on membrane surfaces—Why? The condensate mediated pathway, the amyloid aggregation kinetics, the dominant fibril polymorph formed—may be completely different under the phase separating setting compared to the other two. The differences in the kinetic parameters of the mutants under these two conditions may have nothing to do with mechanisms via the condensate pathway.

We thank the reviewer for this point. We agree that aggregation inside condensates is likely to follow mechanisms that differ from those under shaking/fragmentation or on membrane surfaces. Our aim was not to claim that these conditions reproduce phase separation-driven aggregation, but to use them as a comparison to see whether certain features of the variants consistently affect aggregation. To explain this, we added the following text (page 7 of the main text):

Our final aim was to analyze dispersed-solution amyloid aggregation to better understand why 5-140 α -syn shows distinct behavior compared to the other truncated variants. While aggregation within condensates likely follows different mechanisms from those under shaking or membrane-associated conditions, we used these conventional assays as benchmarks to test whether any of our PTMs consistently influence aggregation.

Minor concerns:

1. Due to the ever-increasing complexity of biomolecular phase separation, it is now more relevant to call it just phase separation (and not phase separation). Especially because we now know that they are often coupled with other phase transition mechanisms such as percolation and micellization (Kar et al. PNAS, 2022).

We assume that the Reviewer is suggesting that we refrain from using the phrase ‘liquid-liquid phase separation’. We have removed all instances of such phrasing from the manuscript.

2. Line 58: “Together, these findings indicate that electrostatic interactions mediate α -syn condensate formation, suggesting a key role for the terminal domains. Furthermore, b-syn, which lacks the hydrophobic eight residue stretch located in the non-amyloid-b component (NAC) domain of α -syn, does not phase separate or aggregate, indicating that this domain is critical for phase separation”. These statements are confusing. Full length a-Syn phase separates at higher Debye screening—making them electrostatically unfavorable (Ray et al. Nat. Chem. 2023; BioRxiv. 2024). The second statement contradicts the first one—the first statement says it is

mediated via the terminal domains via electrostatics and the second statement highlights the importance of the NAC domain. I don't understand what do the authors try to communicate here.

We thank the reviewer for this helpful comment. Our intended meaning is that disruption of electrostatic interactions within the terminal domains promotes α -syn phase separation, but this process also requires the presence of the NAC domain. Thus, both the terminal domains and the NAC domain play important roles in enabling phase separation. In response to this comment, we have revised the manuscript to read (page 2 of the main text):

However, β -syn, which lacks the hydrophobic eight residue stretch located in the non-amyloid- β component (NAC) domain of α -syn, does not phase separate, indicating that this domain is critical for condensate formation. Thus, all three domains are key to determining the likelihood of α -syn phase separation and their precise roles in this process must be examined further.

3. The protein concentrations are written in mM (millimolar) in the main manuscript while in μ M (micromolar in the SI).

We thank the reviewer for noticing this inconsistency. The concentrations have been corrected, and both α -syn and PLK are now reported in μ M throughout the manuscript and Supporting Information.

4. The PLK-aSyn fibrils could be mixed fibrils. The authors should highlight this more prominently in the main manuscript.

We have included an explanation regarding the possibility that PLK and α -syn may form mixed fibrils (page 5 of the main text), as follows:

These amyloids may consist of mixed fibrils in which PLK, which is unlikely to reside in the amyloid core due to its high charge, may bind the negatively charged C-terminal fuzzy coat and promote lateral association, resulting in the thick fibrils we observed.

5. Line 176: "...supporting our earlier hypothesis that condensate growth plateaus when their number falls below a critical threshold". This is not your hypothesis. This is classical condensate growth kinetics via fusion/ripening.

We thank the reviewer for this clarification. We did not intend to present this as our own hypothesis and have removed that sentence.

6. The manuscript has some awkward phrasing that breaks the flow of reading.

We thank the reviewer for their feedback. In response, we have revised portions of the Results and Conclusions sections of the manuscript to improve clarity.

Suggestions to the authors:

Overall, my suggestion would be to calculate the volume fractions and the concentrations of the dense and the dilute phase of condensates formed by the WT and mutant proteins and normalize these parameters to study the effect of surface wetting exclusively.

Also, the authors are encouraged to quantify the concentration of elongation competent amyloid fibrils formed in each case with a classical seeding assay (and not just comment on their number under TEM).

We thank the reviewer for the valuable suggestions. To address them, we have performed additional experiments to estimate the dense- and dilute-phase concentrations and volume fractions of condensates formed by WT and mutant proteins. No significant differences were observed between the protein variants (Supplementary Fig. 26). We have also complemented our microscopy observations with seeded aggregation assays to estimate the concentration of elongation-competent amyloid fibrils under phase separation conditions (Supplementary Fig. 19). We note that fibrils obtained under phase separation conditions exhibit a short lag phase when used as seeds, indicating an activation step prior to steady elongation. We attempted several washing steps to remove residual components from the phase separation preparation, but these did not eliminate the lag. Nevertheless, it is interesting to observe that the elongation data for fibrils obtained under phase separation conditions appears comparable to that for FL fibrils prepared under standard conditions (Fig. 5), indicating a consistent trend in how the truncations affect elongation across the two conditions.

Figure S19. N-terminal truncation delays elongation of α -syn fibrils formed via phase separation.

a/b, Elongation kinetics of FL (blue), 5-140 (red), 11-140 (green) and 19-140 (purple) α -syn fibrils formed under via separation, either in the presence (a) or absence (b) of the corresponding monomeric protein. Three biological repeats are shown. Each repeat is the mean of three technical replicates, semi-transparent error bars represent the standard deviation. c/d, Representative number distribution (c) and the corresponding correlogram (d) of FL (blue), 5-140

(red), 11-140 (green) and 19-140 (purple) α -syn fibrils formed via phase separation and used to induce fibril elongation.

Reviewer #3

This study aims at providing a platform that monitors and quantifies condensate formation, surface wetting, and amyloid aggregation for different alpha-synuclein variants. I believe that trying to systematically establish the relationship amongst these process, if any, is an important question to address. The approach combining 96-well plates with DIC and fluorescence microscopy and Thioflavin-T measurements is technically sound. The authors focus on 4 synuclein variants, WT and 3 which are truncated at the N-terminus. The overall conclusion is that larger truncations, which substantially reduce hydrophobicity, slow down condensate sedimentation, make the condensates more wettable and speeding up amyloid formation. The decoupling between the process of condensate sedimentation, overall slower for these variants, and their ability to aggregate into amyloids is an important finding. The study further highlights the role of the surface to volume ratio in modulating amyloid formation due to the catalytic role of the interface at the condensate surface.

I think this study should be published but I see some limitations which should be addressed to ensure the robustness of the message:

1. The study characterizes 4 variants (including WT). Given that one of them (5-140) has a different behavior from the others, it is hard to draw very robust general conclusions. The authors suggest that changes in hydrophobicity are responsible for the different sedimentation, wettability and aggregation pattern. Can they design a couple of other variants (the more the better), with similar hydrophobicity profiles, and assess their behavior to test their overall hypothesis? This would also further validate the high-throughput feature of their experimental set-up.

We thank the reviewer for this valuable suggestion. In response, we have included analyses for two additional N-terminal modifications with increased hydrophobicity relative to the WT protein: N-terminal acetylation and truncated 14–140 α -syn (Supplementary Table 2). For these variants, we characterized droplet time evolution both in the bulk solution and along the well surface, and we monitored ThT intensity under LLPS conditions. Below, we present representative results, while the complete datasets are provided in Supplementary Fig. 27-29. The addition of these variants has allowed us to refine our hypothesis. Both variants display increased hydrophobicity and reduced N-terminal charge compared to FL α -syn. In both cases, their surface wetting and aggregation rates are enhanced relative to FL, to a similar extent as observed for 11–140 and 19–140 α -syn (Supplementary Fig. 29). These results suggest that charge may instead play an important role in determining condensate wettability.

Representative data from Figure S29. N-terminally modified α -syn condensates modulate surface wetting and aggregation.

a. Representative DIC images acquired manually at the bottom of the well after 20 h incubation of AcFL and 14-140 α -syn in the presence or absence of PLK (scale bars represent 25 μ m). b/c, Raw ThT fluorescence intensity data when AcFL (orange) or 14-140 (pink) α -syn are incubated with 25 μ M PLK in phase separation buffer at 37 $^{\circ}$ C. 60 μ M α -syn alone (grey) is also shown for each variant. Each repeat is the mean of three technical replicates and semi-transparent error bars represent the standard deviation of the mean. d/e, Corresponding normalized ThT intensity data for AcFL (orange) or 14-140 (pink) α -syn incubated with 25 μ M PLK, alongside the FL α -syn data (blue) for comparison. Each repeat is the mean of three technical replicates, error bars represent the standard deviation of the mean.

2. In line with the above, occasionally one single biological replicate is presented (Figure 3). Is this due to extreme variability across biological replicates? Otherwise, the set-up seems ideal to have multiple measurements for one same protein.

We thank the reviewer for this suggestion. We have now included ≥ 3 biological replicates in Figure 3.

3. Data in Figure 5 adds an interesting layer as the author show and discuss the different effects that truncations have on the kinetics of aggregation in the soluble phase, as well as in the stability of the final fibrils. I think these observations are not trivial and would appreciate a sentence on the relationship between kinetics and stability (of condensates and of fibrils) in the discussion.

We thank the reviewer for this suggestion. We have now included a discussion about these findings (page 9 of the main text):

However, our dispersed-solution aggregation analysis indicates that certain N-terminal truncations (i.e., 1–10 and 1–18) yield protofilament-like assemblies, suggesting that these modifications impair both aggregation kinetics and fibril maturation. In contrast, under phase separation conditions, all truncated variants aggregated more rapidly, consistent with enhanced wettability facilitating nucleation. These observations highlight that, although increased nucleation propensity may accelerate aggregation under phase separation conditions, fibril formation and phase separation are governed by distinct mechanisms. Nonetheless, our data suggest that N-terminal truncations bias aggregation toward surface-dependent nucleation pathways.

4. Figure 5, bottom right - y-axis label is missing.

We thank the reviewer for pointing this out. The missing y-axis label in the new version of Figure 5 has been corrected. We have also added secondary nucleation assays to provide a more complete characterization of the aggregation behavior of the protein variants.

- 1 Mesquida, P., Riener, C. K., MacPhee, C. E. & McKendry, R. A. Morphology and mechanical stability of amyloid-like peptide fibrils. *J. Mater. Sci. Mater. Med.* **18**, 1325-1331 (2007).
- 2 Maurstad, G., Prass, M., Serpell, L. C. & Sikorski, P. Dehydration stability of amyloid fibrils studied by AFM. *Eur. Biophys. J.* **38**, 1135-1140 (2009).
- 3 Šneiderienė, G. *et al.* Lipid-induced condensate formation from the Alzheimer's A β peptide triggers amyloid aggregation. *Proc. Natl. Acad. Sci.* **122**, e2401307122 (2025).
- 4 Agarwal, A. *et al.* VAMP2 regulates phase separation of α -synuclein. *Nat. Cell Biol.* **26**, 1296-1308 (2024).
- 5 Ray, S. *et al.* Mass photometric detection and quantification of nanoscale α -synuclein phase separation. *Nat. Chem.* **15**, 1306-1316 (2023).

REVIEWERS' COMMENTS:

Reviewer #1

The authors have satisfactorily addressed my comments, the manuscript is suitable for publication in Communications Chemistry.

Reviewer #2

The authors have done a good job in addressing all my concerns. I have no further comments.

Reviewer #3

The authors did a good job at addressing mine and the other reviewers' points.

We are grateful to the reviewers for their positive final assessments and the constructive feedback provided during the peer-review process, which strengthened the manuscript.